# Multicomponent Synthesis of Unsaturated γ-Lactam Derivatives. Applications as Antiproliferative Agents through the Bioisosterism Approach: Carbonyl vs. Phosphoryl Group

**DOI:** 10.3390/ph15050511

**Published:** 2022-04-22

**Authors:** Xabier del Corte, Adrián López-Francés, Ilia Villate-Beitia, Myriam Sainz-Ramos, Edorta Martínez de Marigorta, Francisco Palacios, Concepción Alonso, Jesús M. de los Santos, José Luis Pedraz, Javier Vicario

**Affiliations:** 1Department of Organic Chemistry I, Faculty of Pharmacy, University of the Basque Country, UPV/EHU Paseo de la Universidad 7, 01006 Vitoria-Gasteiz, Spain; xabier.delcorte@ehu.eus (X.d.C.); adrian.lopez@ehu.eus (A.L.-F.); edorta.martinezdemarigorta@ehu.eus (E.M.d.M.); francisco.palacios@ehu.eus (F.P.); concepcion.alonso@ehu.eus (C.A.); jesus.delossantos@ehu.eus (J.M.d.l.S.); 2NanoBioCel Group, University of the Basque Country (UPV/EHU), 01006 Vitoria-Gasteiz, Spain; aneilia.villate@ehu.eus (I.V.-B.); miriam.sainz@ehu.eus (M.S.-R.); 3Biomedical Research Networking Center in Bioengineering, Biomaterials and Nanomedicine (CIBER-BBN), Faculty of Pharmacy, University of the Basque Country (UPV/EHU), 01006 Vitoria-Gasteiz, Spain; 4Bioaraba, NanoBioCel Research Group, Faculty of Pharmacy, University of the Basque Country (UPV/EHU), 01006 Vitoria-Gasteiz, Spain

**Keywords:** γ-lactams, multicomponent synthesis, antiproliferative activity, bioisosterism, phosphonates, phosphine oxides

## Abstract

We report efficient synthetic methodologies for the preparation of 3-amino and 3-hydroxy 3-pyrrolin-2-ones (unsaturated γ-lactams) through a multicomponent reaction of amines, aldehydes and acetylene or pyruvate derivatives. The densely substituted γ-lactam substrates show in vitro cytotoxicity, inhibiting the growth of the carcinoma human tumor cell lines RKO (human colon epithelial carcinoma), SKOV3 (human ovarian carcinoma) and A549 (carcinomic human alveolar basal epithelial cell). In view of the possibilities for the diversity of the substituents that offer a multicomponent, synthetic methodology, an extensive structure–activity profile is presented. In addition, the bioisosteric replacement of the flat ester group by a tetrahedral phosphonate or phosphine oxide moiety in γ-lactam substrates leads to increased growth inhibition activity. Cell morphology analysis and flow cytometry assays indicate that the main pathway by which our compounds induce cytotoxicity is based on the activation of the intracellular apoptotic mechanism.

## 1. Introduction

Due to a rise in life expectancy and/or decline in the fertility rate, virtually every country in the world is experiencing growth in the number and proportion of older persons in the population. According to data from the United Nations [1], in 2019, 9% of people in the world were over aged age 65. By 2050, it is expected that 16% of the world’s population will be aged over 65, and one in four persons living in Europe and Northern America will be aged 65 or over. Moreover, the number of persons aged 80 years or over is projected to triple, from 143 million in 2019 to 426 million in 2050. These longevity gains are one of the greatest accomplishments of humankind but, at the same time, one of the most formidable challenges for the future with implications for nearly all socioeconomic sectors [2] and an especially high impact in all health care systems worldwide. During the last century, the causes of mortality changed from infectious and parasitic to chronic and degenerative diseases, and accordingly, cancer has become one of the world’s greatest health problems [3]. The systemic treatment of cancer often involves the administration of chemotherapeutic agents, which possess the ability to travel throughout the body and destroy malignant cells [4]. The development of antineoplasic drugs has undergone exponential growth in the last decades, but there is still a serious need to search for newer, safer and more potent cytotoxic drugs, especially due to the known ability of cancer cells to develop resistance to traditional therapies [5,6]. Here, drug discovery plays a crucial role through, first, the identification of drug candidates; second, the synthesis and characterization of target molecules; and finally, the evaluation of their therapeutic efficacy prior to drug development and subsequent clinical trials.

Among the innumerable potential chemotherapeutic agents, the γ-lactam ring (Figure 1) is a key structural scaffold used in medicinal chemistry that is found in the structures of many natural and synthetic bioactive compounds [7]. In particular, 3-pyrrolin-2-ones (Figure 1) are unsaturated γ-lactam derivatives that present a conjugated ring system which possesses latent reactivity for further modifications [8,9,10,11]. Moreover, the structures of these unsaturated γ-lactam derivatives are essential parts of the skeletons of numerous relevant bioactive molecules that show a large variety of biological activities [12,13,14], such as the cytotoxic polyketides Myceliothermophins E, C and D [15], cytotoxic Pukeleumid E present in *Lyngbya majuscule* algae [16], the HIV-integrase inhibitor Oteromicyn [17,18] and the antibiotic Pyrrocidine A [19].

Interestingly, some unsaturated γ-lactam substrates have been identified as p53−MDM2 [20] and STAT3 [21] inhibitors that have strong antiproliferative activity. In addition, several 2-pyrrolidone derivatives have been described as antitumor agents [22,23,24,25,26,27,28,29].

As a part of our ongoing research on the multicomponent synthesis of γ-lactam derivatives, in 2006, we reported that a three-component reaction of amines, aldehydes and pyruvate derivatives in the presence of a Brønsted acid catalyst leads to the formation of 3-amino 3-pyrrolin-2-ones [30]. More recently, we developed an enantioselective version of this reaction [31] and extended this multicomponent protocol to the synthesis of phosphorus- and fluorine-containing isotetronic acid-based γ-lactams [32]. In addition, we were able to demonstrate that these substrates can be prepared through a similar multicomponent reaction using acetylene carboxylates instead of pyruvate derivatives [33]. These multicomponent protocols are considered essential tools in diversity-oriented synthesis [34,35] due to the high degree of molecular diversity achieved and, accordingly, they have become a preferential methodology in the field of medicinal chemistry [36,37]. 3-Amino 3-pyrrolin-2-ones can be seen as cyclic α-dehydro α-amino acids, and such skeletons are known to be present in many bioactive molecules, such as antimicrobials with anti-biofilm activity, caspase-3 inhibitors, analgesics, and antipyretics [38,39,40,41,42]. They also represent the basic structure of dithiopyrrolone antibiotics [43]. In addition, 3-hydroxy 3-pyrrolin-2-ones have been described as HIV integrase inhibitors [44,45], antibacterials [46,47,48], nootropics [49] and antivirals [50]. Additionally they show anticancer activity [26,27].

In this context, very recently, we conducted a study on the antiproliferative activity of 3-amino 3-pyrrolin-2-ones, showing evidence of the ability of these substrates to activate the intracellular apoptotic mechanism [29]. Taking into consideration the potential of five-membered heterocycles containing the 3-pyrrolin-2-one skeleton to act as anticancer agents, we believe that this report on the synthesis of unsaturated 3-hydroxy and 3-amino γ-lactam derivatives obtained by multicomponent methodologies and study of their applications as antiproliferative agents will be of great value to this field.

## 2. Results and Discussion

### 2.1. Chemistry

The multicomponent protocol for the synthesis of 3-amino 3-pyrrolin-2-ones **4** implies the reaction of aromatic amines **1**, aldehydes **2** and acetylene carboxylate derivatives **3** in the presence of a catalytic amount of BINOL-derived phosphoric acid for several hours in refluxing toluene [32]. The presence of MgSO_4_ is necessary in order to remove the water generated in the process. The mechanism of the reaction comprises the initial formation of imine and enamine species **5** and **6** through the reaction of two equivalents of amine substrate **1** with aldehydes **2** and acetylene dicarboxylate derivatives **3**. Next, an acid-promoted Mannich reaction leads to the formation of intermediate **7**, which spontaneously evolves through intramolecular cyclization between the amine and the ester groups to yield γ-lactam substrates **4** (Figure 1).

Following this approach, 13 densely functionalized substrates were synthesized in order to illustrate the synthetic potential of the reaction. First, the reaction was applied to the use of different amines **1** using benzaldehyde (**2a**, R^2^ = Ph) and diethyl acetylenedicarboxylate (**3a**, R^3^ = Et). The reaction with weakly activated *p*-toluidine (**1a**, R^1^ = *p*-CH_3_C_6_H_4_) as an amine substrate provided a very good yield of γ-lactam derivative **4a** after 48 h. (Figure 2, **4a**). However, the use of an electron-rich aniline or aliphatic amines, such as *p*-anisidine (**1b**, R^1^ = *p*-CH_3_OC_6_H_4_) or benzylamine (**1c**, R^1^ = Bn), led to a slight decrease in yield (Figure 2, **4b**,**c**). Remarkably, when electron-deficient amines were used as substrates, the formation of imine and enamine intermediates was initially observed but, in this case, the reaction failed to provide the γ-lactam substrates. It should be noted that, in view of the mechanism proposed for the transformation (Figure 1), the electronic character of the amine substrate might be a crucial factor in the reactivity of the key step of the multicomponent process. Accordingly, while the use of electron-rich amines may benefit the nucleophilic character of enamine species **6**, this would result in a decrease in the electrophilic character of imine species **5**. On the other hand, the use of deactivated amines would result in the activation of imine electrophile **5** and the collateral deactivation of enamine nucleophile **6**. In addition, the slightly lower yield obtained using *p*-anisidine compared to *p*-toluidine can be also attributed to the formation of side products **8a**,**b**, generated by a subsequent nucleophilic addition of an additional molecule of amine to the carboxylate group of lactam **4b**. Amides **8a**,**b** were isolated in 3% and 13% yield from the crude reaction (Figure 2).

The use of other acetylenedicarboxylates **3b**,**c** (R^3^ = *^i^*Pr or Me) as electrophile substrates in the multicomponent reaction with *p*-toluidine (**1a**, R^1^ = *p*-CH_3_C_6_H_4_) and benzaldehyde **2a** (R^2^ = Ph) led to the formation of γ-lactams **4d**,**e** in moderate to good yields (Figure 2). The bulkier di-*iso*-propyl acetylenedicarboxylate **3b** (R^3^ = *^i^*Pr) required a longer reaction time (72 h), which may have also facilitated the formation of the amide side product **8a** as a result of the greater coexistence of *p*-toluidine (**1a**, R^1^ = *p*-CH_3_C_6_H_4_) and γ-lactam **4d**. In the case of dimethyl acetylenedicarboxylate **3c** (R^3^ = Me), amide **8a** formed in equal proportion as with the use of diethyl acetylenedicarboxylate **3a,** even though the reaction proceeded to full conversion with a shorter reaction time (24 h).

Next, the scope of the reaction was extended to the use of different aldehydes by utilizing dimethyl acetylenedicarboxylate **3c** (R^3^ = Me) as the electrophile and *p*-toluidine (**1a**, R^1^ = *p*-CH_3_C_6_H_4_) and benzylamine (**1c**, R^1^ = Bn) as amine substrates. The use of electron-deficient *p*-trifluoromethyl benzaldehyde (**2b**, R^2^ = *p*-CF_3_C_6_H_4_) in the multicomponent reaction provided a more electrophilic imine species **5**, favoring the reactivity of the Mannich intermediate process. Indeed, when *p*-toluidine (**1a**, R^1^ = *p*-CH_3_C_6_H_4_) was used as the reaction partner, a very good yield of lactam **4f** was obtained with no presence of the amide side product, as expected due to the less nucleophilic character of the amine (Figure 1). However, although the reaction with benzylamine (**1c**, R^1^ = Bn) led to the formation of a more nucleophilic enamine intermediate **6** for the Mannich reaction, for this particular case, a low yield of γ-lactam **4g** was obtained, possibly due to the lower electrophilic character found for *N*-benzylimine intermediate **5** (Figure 1).

On the contrary, it was expected that the intermediate Mannich reaction would be disfavored by the use of an aldehyde holding a strong electron-donating substituent, which would generate, in this case, a less electrophilic imine species **5**. Accordingly, the use of *p*-hydroxybenzaldehyde (**2c**, R^2^ = *p*-HOC_6_H_4_) and *p*-toluidine (**1a**, R^1^ = *p*-CH_3_C_6_H_4_) in the multicomponent reaction led to the formation of a modest yield of γ-lactam **4h**, possibly due to the poorer electrophilic character expected for *N*-arylimine intermediate **5** (Figure 1). Nevertheless, the same reaction using benzylamine (**1c**, R^1^ = Bn) produced a good yield of γ-lactam derivative **4i**, where an increase in the nucleophilic character of the enamine intermediate seems to drive the Mannich reaction and overcome the decreased electrophilicity expected for the *N*-benzylimine species **5** (Figure 1). In accordance with the observed reactivity for the multicomponent reaction, *m*-anisaldehyde (**2d**, R^2^ = *m*-MeOC_6_H_4_) provided similar results as benzaldehyde (**2a**, R^2^ = Ph), with a very good yield for *p*-toluidine-derived γ-lactam **4j** and a slightly lower yield for benzylamine-derived γ-lactam **4k** (Figure 2). Finally, the scope of the reaction was completed using a disubstituted benzaldehyde. In this case, by following the same multicomponent protocol with dimethyl acetylenedicarboxylate **3c** (R^3^ = Me) and vanillin (**2e**, R^2^ = 3-MeO-4-HO-C_6_H_3_), γ-lactams **4l**–**m** were obtained with moderate to good yields (Figure 2). In addition, the benzyl group at the enamine moiety in **4c** was selectively removed by hydrogenolysis through treatment with EtOH under an H_2_ atmosphere at 80 psi in the presence of 10% mol of palladium on carbon, leading to the production of lactam derivative **9** with a quantitative yield (Figure 2). Remarkably, under those conditions, only one of the benzyl groups was removed, and the carbon–carbon double bond remained intact.

Due to their chemical similitude to natural phosphate metabolites, phosphonate derivatives show multiple biological activities, and for this reason, they have numerous applications in medicine and agrochemistry [51,52,53,54,55,56]. Thus, in order to further broaden the scope of the Brønsted acid-catalyzed multicomponent reaction, we tried to extend the synthetic protocol to the use of activated-acetylene-bearing phosphorated groups (substitution of a carboxylate by a phosphonate group). However, the reaction using methyl 3-(diethoxyphosphinyl)-2-propynoate (MeO_2_C-C≡C-P(O)(OEt)_2_) instead of dialkyl acetylenedicarboxylates gave complex mixtures. Nevertheless, the use of phosphorus-substituted pyruvates as surrogates of acetylenedicarboxylates proved to be an excellent choice. To our delight, the Brønsted-acid-catalyzed multicomponent reaction of amines **1**, aldehydes **2** and phosphorated pyruvates **10** in refluxing MTBE led to the formation of an excellent yield of phosphorus-substituted γ-lactam derivatives **12** after 48 h. (Figure 3). Similar to the multicomponent reaction using acetylene carboxylates (Figure 1, vide supra), the enamine intermediate **6** was generated in this case through an amine-carbonyl condensation reaction between pyruvate derivative **10** and amine **1** using MgSO_4_ to remove the water released. Next, 3-amino 3-pyrrolin-2-ones **11** were formed in an identical Mannich reaction. Due to the high steric hindrance expected in the highly functionalized heterocycle, enamine derivatives **11** are not isolable, and 3-hydroxy 3-pyrrolin-2-ones **12** were obtained in this case after spontaneous hydrolysis of the enamine moiety (Figure 3).

First, the reaction was applied to different amine substrates **1** using benzaldehyde **2a** (R^2^ = Ph) and diethylphosphoryl substituted pyruvate **10a** (R^3^ = OEt), providing good to excellent yields of γ-lactam derivatives **12a**–**c** (Figure 3). The scope of the reaction was extended to the use of different aldehydes using diethylphosphoryl-substituted pyruvate **10a** (R^1^ = OEt) and *p*-toluidine (**1a**, R^1^ = *p*-CH_3_C_6_H_4_) as the reaction partners and providing good to excellent yields of γ-lactam derivatives **12d**–**h** (Figure 3). In addition, the multicomponent reaction was applied to diverse phosphorus-substituted pyruvates to produce several different phosphonate and phosphine oxide-substituted lactams **12i**–**p** (Figure 3).

Next, in order to extend the structural diversity obtained in this study, some additional 3-hydroxy 3-pyrrolin-2-ones **13** were prepared from alkyloxycarbonyl-substituted 3-amino 3-pyrrolin-2-ones **4** previously obtained from the multicomponent reaction using acetylenedicarboxylates. Upon treatment of γ-lactams **4** under acidic conditions, selective hydrolysis of the enamine moiety was observed, leading to the formation of enol-containing lactam substrates **13a–g**. The reaction was applied successfully to ethyl ester-substituted γ-lactam substrates **13a**–**c** derived from *p*-toluidine (**1a**, R^1^ = *p*-CH_3_C_6_H_4_), *p*-anisidine (**1b**, R^1^ = *p*-CH_3_OC_6_H_4_) or benzylamine (**1c**, R^1^ = Bn) and could be also extended to methyl esters derived from a variety of aldehydes for the preparation of γ-lactams **13d**–**g** (Figure 4).

With this collection of highly functionalized γ-lactam derivatives in our hands, we studied their biological activity. The antiproliferative activity of the substrates against several cancer cell lines was investigated.

### 2.2. Biological Results

The in vitro cytotoxicity of the γ-lactam derivatives was evaluated by testing their antiproliferative activity against several human cancer cell lines. The cell counting kit (CCK-8) assay was used for the evaluation of growth inhibition. Moreover, nonmalignant MRC5 lung fibroblasts were tested to study the selective toxicity [57], and chemotherapeutic doxorubicin was used as a reference.

In the first study, we tested the cytotoxicity of 3-amino 3-pyrrolin-2-ones **4**, **8** and **9** obtained from SKOV3 (human ovarian carcinoma) and A549 (carcinomic human alveolar basal epithelial cell) cell lines. The cell proliferation inhibitory activity of the γ-lactams is shown as IC_50_ values (Table 1).

Accordingly, γ-lactam derivative **4a**, derived from *p*-toluidine (**1a**), benzaldehyde (**2a**) and diethyl acetylenedicarboxylate (**3a**) showed a modest IC_50_ value of 11.70 ± 1.02 µM against the A549 cell line (Table 1, Entry 1). Analogous γ-lactams **4b**,**c**, derived from *p*-anisidine or benzylamine presented IC_50_ values of 14.26 ± 1.80 and 2.42 ± 0.15 µM (Table 1, Entries 2–3). Remarkably, substrates **4a**–**c** did not show significant cytotoxicity against SKOV3, while they displayed very good selectivity against nonmalignant cells with IC_50_ values higher than 50 µM (Table 1, Entries 1–3). Switching the ethyl ester with an *iso*-propylester group at the 5-membered ring resulted in an improved level of toxicity towards the A549 and SKOV3 cell lines with IC_50_ values of 3.34 ± 0.29 and 48.45 ± 2.90 µM, respectively, in substrate **4d** while maintaining good selectivity with respect to the MRC5 cell line (Table 1, Entry 4). Remarkably, the presence of a methyl ester substituent in **4e** provided a notable improvement in the inhibition of cell growth in the A549 cell line with a very good IC_50_ value of 1.67 ± 0.49 µM and very good selectivity against the SKOV3 and MRC5 cell lines (Table 1, Entry 5).

With the view that the best toxicity level was obtained for methyl ester derivative **4e**, we next extended the structure–activity relationship study to an investigation of influence of the substituent at position 5 of the γ-lactam ring using **4e** as the model. Although the effect of the introduction of fluorine atoms into the structure of organic compounds is rather difficult to predict, it very often leads to increased activity [58,59,60,61]. The key properties that make fluorine-containing compounds attractive in chemical biology include the small atomic radius and high electronegativity of the fluorine atom and the low polarizability of the C–F bond. In addition, the fact that the only natural isotope of ^19^F atom has a nuclear spin of ½ makes it ideal for monitoring studies by NMR. For this reason, we next tested the in vitro cytotoxicity of trifluoromethyl-containing γ-lactams **4f**,**g**. The introduction of a *para*-trifluorophenyl substituent at the 5-membered ring did not result in improved activity, and IC_50_ values of 42.58 ± 2.55 and 7.64 ± 0.17 µM were obtained in A549 cell line for compounds **4f**,**g**, respectively. However, compound **4f** did show some toxicity against the SKOV3 cell line with a moderate IC_50_ value of 30.27 ± 1.03 µM. Remarkably, both compounds exhibited very high selectivity towards malignant cells with IC_50_ values higher than 50 µM in the MRC5 cell line. (Table 1, Entries 6–7).

The antioxidant properties of phenols are known to be associated with the antitumor activities of a plethora of compounds bearing this moiety [62]. Accordingly, the antiproliferative activity of phenol-substituted γ-lactams **4h**,**i**, was tested. Indeed, an excellent IC_50_ value of 1.98 ± 0.18 µM was found for the A549 cell line for *p*-toluidine-derived γ-lactam **4h**. Although compound **4h** also showed some toxicity against nonmalignant cells, its selectivity was found to be 5 times higher compared with A549 cells. Likewise, in this case, the toxicity towards ovarian carcinoma was comparable to that observed in the MRC5 cell line (Table 1, Entry 8). In addition, benzylamine-derived γ-lactam **4i** presented IC_50_ values of 10.71 ± 1.35 and 21.91 ± 1.53 µM in the A549 and SKOV3 cell lines, respectively, although a similar level of toxicity was found for nonmalignant cells (Table 1, Entry 9).

The methoxy group is a strong electron-donating substituent in aromatic rings that is known to be a widespread motif in drugs and natural products. The introduction of this moiety to potential anticancer agents very often leads to increased selective activity [63,64], which is attributed in part to its weak to medium antimitotic activity. Consequently, the cytotoxicity of *m*-methoxyphenyl-substituted γ-lactams **4j**,**k** was next explored. For the particular case of *p*-toluidine-derived γ-lactam **4j**, some cytotoxicity was observed against the A549 and SKOV3 cell lines with IC_50_ values of 13.03 ± 1.48 and 43.93 ± 1.66 µM, respectively, although not much selectivity was obtained compared with nonmalignant cells (Table 1, Entry 10). Moreover, switching *p*-toluidine with a benzylamine group in **4k** did not have a positive effect on the cytotoxicity against the A549 cell line with an IC_50_ value of 11.39 ± 1.49 µM. However, compound **4k** was found to be very selective in A549 cells compared with the SKOV3 or MRC5 cell lines (Table 1, Entry 11). To our surprise, the combination of phenol and methoxy moieties in γ-lactams **4l**,**m** provided excellent IC_50_ values of 0.11 ± 0.016 and 6.02 ± 1.01 µM in the A549 cell line for *p*-toluidine- and benzylamine-derived substrates, respectively, with a high level of selectivity towards nonmalignant cells. Noticeably, *p*-toluidine derivative **4l** delivered a very good IC_50_ value of 1.23 ± 0.31 µM against SKOV3 cells (Table 1, Entries 12–13).

In addition, the effect of the replacement of the methyl ester with an amide group had disparate effects on the antiproliferative activity of γ-lactam substrates. While a very good IC_50_ value of 2.97 ± 0.29 µM was obtained for the A549 cell line for *p*-toluidine derivative **8a**, a modest IC_50_ value of 32.38 ± 1.58 µM was observed for *p*-anisidine derived substrate **8b**. Further, compounds **8a**,**b** also showed toxicity in the SKOV3 cell line with IC_50_ values of 6.95 ± 0.59 and 16.62 ± 0.19 µM, respectively. In addition, compound **8a** was found to be very selective towards malignant cells, although substrate **8b** presented significant toxicity in the MRC5 cell line (Table 1, Entries 14–15). Likewise, *N*-debenzylated substrate **9** gave slightly worse values compared with its precursor, γ-lactam **4c** (Table 1, Entry 3 vs. Entry 16).

In order to further extend our structure–activity study, we next tested the antiproliferative activity of the 3-hydroxy γ-lactam derivatives **13** obtained from the hydrolysis of their parent enamine derivatives (see Figure 4). The replacement of enamine with an enol moiety in ethyl ester substituted structures **4a**–**c** resulted in similar or slighter lower cytotoxic activity against the A549 cell line in compounds **13a**–**c** (Table 2, Entries 1–3 vs. Table 1, Entries 1–3), showing IC_50_ values of 15.73 ± 1.27, 13.05 ± 0.56 and 4.50 ± 0.18 µM, respectively. Moreover, compounds **13a**–**c** did not present significant toxicity in the SKOV3 or MRC5 cell lines. Similarly, methyl-ester-substituted 3-hydroxy γ-lactam **13d** holding a *p*-trifluoromethyl substituent at the chiral carbon of the lactam ring presented lower toxicity against the A549 cell line compared with the parent enamine substrate **4g**, with an IC_50_ value of 19.13 ± 3.00 µM and no toxicity towards the SKOV3 and MRC5 cell lines (Table 2, Entry 4 vs. Table 1, Entry 7). The same lowering of antiproliferative activity was observed in *m*-anisyl derivatives **13e**,**f** and vanillin derivative **13g** relative to their enamine precursors **4j**–**l**, with IC_50_ values of 17.64 ± 3.76, 15.96 ± 1.97 and 13.30 ± 2.19 µM, respectively (Table 2, Entries 5–7 vs. Table 1, Entries 10–12). Compounds **13d**,**e** showed no toxicity in the SKOV3 and MRC5 cell lines.

Bioisosterism represents an approach that is widely used for the rational modification of lead compounds into safer and more clinically effective agents [65]. Accordingly, it is well known that the substitution of a carboxylate with a phosphonate group in active substrates may result in new or increased activity [51,52,53,54,55,56]. For this reason, we next studied the antiproliferative activity of phosphorus-substituted γ-lactam derivatives **12** against the A549 and SKOV3 cell lines (Table 3). Indeed, the replacement of the ethyl carboxylate group with a diethyl phosphonate substituent in *p*-toluidine- and *p*-anisidine-derived lactams **13a**,**b**, resulted in an increase in the cytotoxic activity towards the A549 cell line in **12a**,**b** with IC_50_ values of 3.11 ± 0.31 and 4.56 ± 0,44 µM, respectively, and a high level of selectivity compared with the SKOV3 and MCR5 cell lines (Table 3, Entries 1–2 vs. Table 2, Entries 1–2). However, phosphorated γ-lactam **12c** derived from *o*-fluoroaniline presented decreased activity in the A549 cell line with an IC_50_ value of 16.03 ± 1.49 µM.

With these results in hand, we next studied the effect of the substitution at the chiral carbon of the five membered ring 3-hydroxy 3-pyrrolin-2-ones **12d**–**h** using the most active substrate **12a**, derived from *p*-toluidine and diethyl phosphonate, as a model. The introduction of a strong electron-withdrawing *p*-nitrophenyl group at the stereogenic carbon of the γ-lactam ring had a very negative effect on the cytotoxicity, and IC_50_ values higher than 50 μM were found for compound **12d** in both the A549 and SKOV3 cell lines (Table 3, Entry 4).

In spite of the benefits expected from the introduction of a fluorine atom into the heterocyclic structure [58,59,60,61], *p*-fluorophenyl-substituted lactam **12e** provided a slightly worse IC_50_ value of 6.6 ± 0.58 μM in the A549 cell line relative to the parent compound **12a** (Table 3, Entry 5 vs. Entry 1). The introduction of other heteroaromatic, ester or aliphatic substituents also had a negative effect on the toxicity of substrates **12**. Accordingly, a drop in antiproliferative activity towards the A549 cell line was observed for 2-thienyl, ethoxycarbonyl and *iso*-propyl substituted γ-lactams **12f**–**h** with IC_50_ values of 23.29 ± 2.4, 8.27 ± 0.91 and 24.20 ± 0.81 μM, respectively. However, compounds **12f**–**h** presented high selectivity compared with the SKOV3 cell line and nonmalignant cells (Table 3, Entries 6–8).

Continuing with our interest on phosphorus-containing heterocycles, we next tested the cell growth inhibition activity of bulkier di-*iso*-propyl phosphonates **12i**–**k**. Although slightly higher IC_50_ values of 5.36 ± 0.28 and 5.91 ± 0.69 μM were found in the A549 cell line for phenyl- and *p*-fluorophenyl-substituted lactams **12i**,**j** with respect to their parent diethyl phosphonate derivatives **12a**,**e**, those substrates presented significant toxicity in SKOV3 cells with IC_50_ values of 11.56 ± 3.36 and 15.55 ± 1.60 μM and high selectivity towards nonmalignant cells (Table 3, Entries 9, 10 vs. Entries 1, 5). Moreover, the presence of a carboxylate group in di-*iso*-propyl phosphonate substituted γ-lactam **12f** led to a complete loss of cytotoxicity in all cancer cell lines (Table 3, Entry 11).

Although the appearance of phosphine oxides in drug discovery is rare compared with their counterparts phosphates, phosphonates or phosphoramidates, a few of these derivatives have been proven to be excellent drug candidates, such as the anticancer drugs ridaforolimus [66,67] or brigatinib [68,69]. For this reason, in order to further expand the study of the effect of the phosphorus substituent at the γ-lactam ring, we extended the structure–activity study to substrates **12l**–**p**, holding a phosphine oxide substituent at C-4. Accordingly, first, the antiproliferative activity of 5-phenyl-substituted lactam substrates was studied, holding a diphenylphosphine oxide moiety at C-4. In this case, *p*-toluidine-derived substrate **12l** presented lower cytotoxicity than its phosphonate analogs **12a**,**i** with an IC_50_ value of 11.86 ± 1.35 μM in the A549 cell line (Table 3, Entry 12 vs. Entries 1, 9). However, the presence of a diphenylphosphine oxide moiety in *p*-anisidine and *o*-fluoroaniline derivatives **12m**,**n** resulted in an improvement in cytotoxicity towards the A549 cell line compared with the parent diethyl phosphonate substrates **12b**,**c**, showing IC_50_ values of 3.72 ± 0.32 and 5.5 ± 1.35 μM, respectively (Table 3, Entries 13, 14 vs. Entries 2, 3). Likewise, the presence of a *p*-fluorophenyl substituent at the chiral center of phosphine oxide derivative **12o** provided a very active substrate with IC_50_ values of 1.46 ± 0.19 and 21.97 ± 3.42 μM in the A549 and SKOV3 cell lines, respectively (Table 3, Entry 15). Nevertheless, the parent perfluorophenyl derivative **12p** showed less activity with a modest IC_50_ value of 20.34 ± 0.79 μM against the A549 cell line and no toxicity in the SKOV3 cell line (Table 3, Entry 16). It should be remarked that all active phosphorated 3-hydroxy 3-pyrrolin-2-ones **12** presented excellent selectivity towards malignant cells with IC_50_ values higher than 50 μM in nonmalignant cells in all cases.

Finally, the most active and selective γ-lactam substrates were chosen, and their cytotoxicity levels were tested against the RKO cell line (human colon epithelial carcinoma). Although most of the substrates were found to be inactive towards this cell line, compounds **8a** and **12l** showed significant grown inhibition activity with IC_50_ values of 18.67 ± 1.31 and 33.62 ± 0.41 μM, respectively (See Appendix A).

In order to clarify the mechanism of action of these unsaturated γ-lactam derivatives, cell morphology analysis and flow cytometry assays were performed (See Appendix A). First, flow cytometry assays were conducted using compounds **4l** (1 µM) and **12a** (5 µM) in A549 cells, and measurements were carried out 24 h post-exposure. This assay allowed us to distinguish four separate cell populations: live cells (FL-1 and FL-3 negative), early apoptotic live cells (FL-1 positive, FL-3 negative), late apoptotic dead cells (FL-1 and FL-3 positive) and nonapoptotic dead cells (FL-1 negative, FL-3 positive). At the time of measurement, around 80% of cells exposed to compound **4l** showed a positive FL-1 signal and negative FL-3 signal, indicating an early apoptotic cell population, while in cells exposed to compound **12a**, the percentage of early apoptotic cells was lower, around 50% (Figure 5). At this time point, percentages of late apoptotic cells were still low in both cases with values below 10%. These results indicate that, within 24 h after exposure to compounds **4l** and **12a**, around 90% and 60% of cells, respectively, had activated the apoptotic mechanisms. Necrotic cells that were nonapoptotic represented less than 1% of the whole cell population for both compounds, suggesting that the majority of dead cells detected in this study were apoptotic cells. Finally, nonapoptotic live cells represented around 10% of the whole cell population. These results reveal that compound **4l** has a higher capacity to induce apoptosis compared with **12a**. Therefore, further cell imaging studies were carried out with the more active compound **4l**.

The cell morphology of the A549 cell line was analyzed at different exposure times after the addition of three different concentrations of 3-amino 3-pyrrolin-2-one **4l** to observe the cellular modifications during the treatment. The first studied concentration was 1.1 µM, which is a ten-fold higher concentration compared with the IC_50_ of **4l**. At that high concentration, a disruption of cellular growth was noticed, especially after 6 h of exposure, and the suspension of cellular growth became gradually more pronounced over time (Figure 6A). Then, the IC_50_ concentration of **4l** was used (0.11 µM) for this experiment, and peak cellular death was observed 24 or 48 h after the addition of the compound. Then, live cells started growing again, and higher cellular confluence was witnessed after 72 h (Figure 6B). On the contrary, the addition of **4l** at 0.02 µM, a five-fold lower concentration than the measured IC_50_, showed cells with a healthy, uniform morphology, and cellular growth was recognizable over time, meaning that this concentration was well tolerated (Figure 6C).

In addition, it is worth mentioning that all compounds, with the exception of 3-hydroxy γ-lactam derivative **12p**, fulfilled the requirements for orally active drugs for use in humans in accordance with Lipinski’s rule of five. In addition, and according to the predictions, most of the synthesized compounds may have high levels of gastrointestinal absorption, however not all of them seem to have the ability to cross the blood–brain barrier (see Appendix A).

## 3. Material and Methods

### 3.1. Chemistry

#### 3.1.1. General Experimental Information

Solvents for extraction and chromatography were of technical grade. All solvents used in reactions were freshly distilled from appropriate drying agents before use. All other reagents were recrystallized or distilled as necessary. All reactions were performed under an atmosphere of dry nitrogen. Analytical TLC was performed with silica gel 60 F_254_ plates. Visualization was accomplished by UV light. ^1^H, ^13^C, ^31^P and ^19^F NMR spectra were recorded on a Varian Unity Plus (Varian Inc., NMR Systems, Palo Alto, CA, USA) (at 300 MHz, 75 MHz, 120 MHz and 282 MHz, respectively) and on a Bruker Avance 400 (Bruker BioSpin GmbH, Rheinstetten, Germany) (at 400 MHz for ^1^H, and 100 MHz for ^13^C). Chemical shifts (δ) are reported in ppm relative to the residual CHCl_3_ (δ = 7.26 ppm for ^1^H and δ = 77.16 ppm for ^13^C NMR), and phosphoric acid (50%) was used as an external reference (δ = 0.0 ppm) for ^31^P NMR spectra. Coupling constants *(J)* are reported in Hertz. Data for the ^1^H NMR spectra are reported as follows: chemical shift, multiplicity, coupling constant, and integration. The multiplicity abbreviations used are as follows: s = singlet, d = doublet, t = triplet, q = quartet, m = multiplet). ^13^C NMR peak assignments were supported by Distortionless Enhanced Polarization Transfer (DEPT). High resolution mass spectra (HRMS) were obtained by positive-ion electrospray ionization (ESI). Data are reported in the form *m*/*z* (intensity relative to base = 100). Infrared spectra (IR) were produced with a Nicolet iS10 Termo Scientific spectrometer (Thermo Scientific Inc., Waltham, MA, USA) as neat solids. Peaks are reported in cm^−1^.

#### 3.1.2. Compound Purity Analysis

All synthesized compounds were analyzed by HPLC to determine their purity. The analyses were performed with an Agilent 1260 infinity HPLC system (Agilent, Santa Clara, CA, USA) (C-18 column, Hypersil, BDS, 5 μm, 0.4 mm × 25 mm) at room temperature. All tested compounds were dissolved in dichloromethane, and 5 μL of the sample was loaded onto the column. Ethanol:Heptane (90:10) or Ethanol:Ethyl acetated (50:50) was used as the mobile phase, and the flow rate was set at 1.0 mL/min. The maximal absorbance at the range of 190–400 nm was used as the detection wavelength. The purity of all tested lactam derivatives **4**, **8**, **9**, **12** and **13** was >95%, which meets the purity requirement of the Journal.

#### 3.1.3. Representative Experimental Procedures and Characterization Data for Compounds **4**, **8**, **9**, **12** and **13**

##### Representative Procedure for the Multicomponent Reaction of Amines **1**, Aldehydes **2** and Acetylenes **3**

A solution of amine **1** (4 mmol), aldehyde **2** (2 mmol), acetylene dicarboxylate derivative **3** (2 mmol) and BINOL-derived phosphoric acid (70 mg, 0.2 mmol) was stirred in the presence of anhydrous MgSO_4_ in Toluene (10 mL) at 110 °C for 24–72 h (see ESI). Then, the volatiles were distilled off at reduced pressure, and the crude residue was purified by column chromatography (Hexanes/AcOEt) to produce pure γ-lactams **4** and **8**.

*Ethyl 5-oxo-2-phenyl-1-(p-tolyl)-4-(p-tolylamino)-2,5-dihydro-1H-pyrrole-3-carboxylate (**4a**).* The general procedure was followed using *p*-toluidine (**1a**) (429 mg, 4 mmol), benzaldehyde (**2a**) (204 μL, 2 mmol) and diethyl acetylenedicarboxylate (**3a**) (320 μL, 2 mmol). The residue was purified by column chromatography (Hexanes/AcOEt 8:2), producing 656 mg (77%) of **4a** as a white solid. M.p. (Et_2_O) 154–155 °C. ^1^H NMR (400 MHz, CDCl_3_): δ 8.17 (bs, 1H, NH), 7.34 (d, ^3^*J_HH_* = 8.5 Hz, 2H, 2 × CHar), 7.26–7.21 (m, 5H, 5 × CHar), 7.12 (d, ^3^*J_HH_* = 8.3 Hz, 2H, 2 × CHar), 7.08 (d, *J* = 8.5 Hz, 2H, 2 × CHar), 7.03 (d, ^3^*J_HH_* = 8.3 Hz, 2H, 2 × CHar), 5.77 (s, 1H, CHN), 4.01 (q, ^3^*J_HH_* = 7.1 Hz, 2H, CH_2_ OEt), 2.33 (s, 3H, CH_3_ Tol), 2.23 (s, 3H, CH_3_ Tol), 1.01 (dd, ^3^*J_HH_* = 7.1 Hz, ^3^*J_HH_* = 7.1 Hz, 3H, CH_3_ OEt). ^13^C {^1^H} NMR (101 MHz, CDCl_3_) δ 164.7 (C=O ester), 164.1 (C=O amide), 142.7 (=C_quat_), 137.2 (C_quat_), 136.1 (C_quat_), 135.5 (C_quat_), 134.6 (C_quat_), 134.2 (C_quat_), 129.5 (2 × CHar), 129.1 (2 × CHar), 128.4 (2 × CHar), 128.1 (CHar), 127.83(2 × CHar), 123.2 (2 × CHar), 122.8 (2 × CHar), 108.9 (=C_quat_), 63.3 (CHN), 60.2 (CH_2_ OEt), 21.1 (CH_3_), 21.0 (CH_3_), 14.0 (CH_3_ OEt). FTIR (neat) ν_max_: 3289 (N-H), 1701 (C=O), 1679 (C=O), 1632 (C=C). HRMS (Q-TOF) *m*/*z* calcd for C_27_H_26_N_2_O_3_ [M]^+^ 426.1943, found 426.1950.

*5-Oxo-2-phenyl-N,1-di-p-tolyl-4-(p-tolylamino)-2,5-dihydro-1H-pyrrole-3-carboxamide (**8a**).* The general procedure was followed using *p*-toluidine (**1a**) (429 mg, 4 mmol), benzaldehyde (**2a**) (204 μL, 2 mmol) and diethyl acetylenedicarboxylate (**3a**) (284 mg, 2 mmol). The residue was purified by column chromatography (Hexanes/AcOEt 8:2), producing 29 mg (3%) of **8a** as a white solid. A higher yield (22%) was achieved using di-*iso*-propyl acetylenedicarboxylate (**3b**) (396 mg, 2 mmol). M.p. (Et_2_O) 226 °C (dec.). ^1^H NMR (300 MHz, CDCl_3_) δ 8.31 (bs, 1H, NH), 7.38–7.28 (m, 6H, 6 × CHar), 7.11–7.04 (m, 7H, 7 × CHar), 6.96 (d, ^3^*J_HH_* = 8.5 Hz, 2H, 2 × CHar), 6.84 (d, ^3^*J_HH_* = 8.5 Hz, 2H, 2 × CHar), 6.63 (bs, 1H, NH), 5.85 (s, 1H, CHN), 2.28 (s, 3H,CH_3_ Tol), 2.25 (s, 3H, CH_3_ Tol), 2.24 (s, 3H, CH_3_ Tol). ^13^C {^1^H} NMR (75 MHz, CDCl_3_) (75 MHz, CDCl_3_) δ 164.75 (C=O), 162.12 (C=O), 139.1 (=C_quat_), 136.6 (C_quat_), 136.1 (C_quat_), 135.8 (C_quat_), 134.8 (C_quat_), 134.6 (C_quat_), 133.9 (C_quat_), 133.8 (C_quat_), 129.7 (4 × CHar), 129.5 (2 × CHar), 129.4 (2 × CHar), 129.3 (CHar), 128.0 (2 × CHar), 123.3 (2 × CHar), 122.5 (2 × CHar), 119.8 (2 × CHar), 112.4 (=C_quat_), 63.8 (CHN), 21.1 (CH_3_), 21.0 (CH_3_), 21.0 (CH_3_). FTIR (neat) ν_max_: 3309 (N-H), 3251 (N-H), 1685 (C=O), 1632 (C=C). HRMS (Q-TOF) *m*/*z* calcd for C_27_H_26_N_2_O_3_ [M]^+^ 487.22598, found 487.2255.

##### Representative Procedure for the Multicomponent Reaction of Amines **1**, Aldehydes **2** and Phosphorated Pyruvate Derivatives **10**

A solution of amine **1** (4 mmol), aldehyde **2** (2 mmol), ethyl pyruvate derivative **10** (6 mmol), BINOL-derived phosphoric acid (70 mg, 0.2 mmol) and anhydrous MgSO_4_ was stirred in MTBE (10 mL) at 55 °C for 48 h. The volatiles were distilled off at reduced pressure and the crude residue was purified by column chromatography (Hexanes/AcOEt) to produce pure γ-lactams **12**.

*Diethyl (4-hydroxy-5-oxo-2-phenyl-1-(p-tolyl)-2,5-dihydro-1H-pyrrol-3-yl)phosphonate (**12a**).* The general procedure was followed using *p*-toluidine (**1a**) (429 mg, 4 mmol), benzaldehyde (**2a**) (204 μL, 2 mmol) and ethyl 3-(diethoxyphosphoryl)-2-oxopropanoate (**10a**) (1.513 g, 6 mmol). The residue was purified by column chromatography (Hexanes/AcOEt 3:7), producing 770 mg (96%) of **12a** as a white solid. M.p. (Et_2_O) 145–147 °C. ^1^H NMR (300 MHz, CDCl_3_) δ 9.75 (bs, 1H, OH), 7.36 (d, ^3^*J*_HH_ = 8.5 Hz, 2H, 2 × CHar), 7.26–7.13 (m, 5H, 5 × CHar), 7.05 (d, ^3^*J*_HH_ = 8.3 Hz, 2H, 2 × CHar), 5.58 (d, ^3^*J*_PH_ = 2.8 Hz, 1H, CH), 4.15 (m, 2H, OCH_2_), 3.70 (m, 1H, OCH_2_), 3.16 (m, 1H, OCH_2_), 2.23 (s, 3H, CH_3_, Tol), 1.39 (dd, ^3^*J*_HH_ = 7.1 Hz, ^3^*J*_HH_ = 7.1 Hz, 3H, OEt), 0.80 (dd, ^3^*J*_HH_ = 7.1 Hz, ^3^*J*_HH_ = 7.1 Hz, 3H, OEt). ^13^C NMR {^1^H} (75 MHz, CDCl_3_) δ 163.0 (d, ^2^*J*_PC_ = 19.8 Hz = C_quat_), 160.0 (d, ^3^*J*_PC_ = 6.5 Hz, C=O), 135.7 (C_quat_), 135.5 (C_quat_), 134.1 (C_quat_), 129.7 (2 × CHar), 129.1 (2 × CHar), 128.7 (CHar), 127.1 (2 × CHar), 121.9 (2 × CHar), 106.6 (d, ^1^*J*_PC_ = 200.8 Hz, C_quat_-P), 62.98 (d, ^2^*J*_PC_ = 5.4 Hz, CH_2_O), 62.71 (CH), 62.56 (d, ^2^*J*_PC_ = 4.5 Hz, CH_2_O), 21.0 (CH_3_), 16.5 (d, ^3^*J*_PC_ = 6.4 Hz, CH_3_), 15.6 (d, ^3^*J*_PC_ = 7.5 Hz, CH_3_). ^31^P NMR (121 MHz, CDCl_3_) δ 15.4. FTIR (neat) ν_max_: 3112 (OH), 1692 (C=O), 1663 (C=C), 1165 (P=O), 1021 (P-O-C). HRMS (ESI-TOF) *m*/*z*: [M+H]^+^ calcd for C_21_H_25_NO_5_P 402.1470; Found 402.1468.

##### Representative Procedure for the Hydrolysis of 3-Amino 3-Pyrrolin-2-Ones **4**: Synthesis of 3-Hydroxy 3-Pyrrolin-2-Ones **13**

Compound **4** (0.5 mmol) was dissolved in a 1:1 mixture of a 3 M aqueous solution of HCl and THF (10 mL), and the reaction was heated to 66 °C overnight. The reaction progress was monitored by TLC.(Scharlab, S.L, Barcelona, Spain) The resulting mixture was concentrated under reduced pressure in order to evaporate THF and then diluted with AcOEt (10 mL). The organic layer was washed with a 3M aqueous solution of NaOH (2 × 5 mL) and H_2_O (2 × 5 mL) and dried over anhydrous MgSO_4_, and the crude residue was purified by crystallization from Et_2_O/pentane to produce pure γ-lactams **13**.

*Ethyl 4-hydroxy-5-oxo-2-phenyl-1-(p-tolyl)-2,5-dihydro-1H-pyrrole-3-carboxylate (**13a**).* The general procedure was followed, producing 0.161 g (95%) of **13a** as a white solid. M.p. (Et_2_O) 170–172 °C. ^1^H-NMR (300 MHz, CDCl_3_) δ 9.19 (bs, 1H, OH), 7.38 (d, ^3^*J_HH_* = 8.2 Hz, 2H, 2 × CHar), 7.32–7.25 (m, 5H, 5 × CHar), 7.09 (d, ^3^*J_HH_* = 8.2 Hz, 2H, 2 × CHar), 5.74 (s, 1H, CH), 4.20 (q, ^3^*J_HH_* =, 7.1 Hz, 2H, CH_2_O), 2.26 (s, 3H, CH_3_ Tol), 1.20 (dd, ^3^*J_HH_* = 7.1Hz, ^3^*J_HH_* = 7.1Hz, 3H, CH_3_ OEt). ^13^C {^1^H} NMR (75 MHz, CDCl_3_) δ 165.0 (C=O ester), 162.9 (C=O amide), 156.4 (=C_quat_), 135.7 (C_quat_), 135.3 (C_quat_), 133.7 (C_quat_), 129.6 (2 × CHar), 128.6 (2 × CHar), 128.5 (CHar), 127.6 (2 × CHar), 122.4 (2 × CHar), 113.1 (=C_quat_), 61.8 (CHN), 61.2 (CH_2_ OEt), 20.9 (CH_3_ Tol), 14.0 (CH_3_ OEt). FTIR (neat) ν_max_: 3425 (O–H), 1704 (C=O), 1675 (C=O), 1643 (C=C). HRMS (Q-TOF) *m*/*z* calculated for C_27_H_26_N_2_O_3_ [M]^+^ 337.1314, found 337.1319.

### 3.2. Biology

#### 3.2.1. Materials

Reagents and solvents were used as purchased without further purification. All stock solutions of the investigated compounds were prepared by dissolving the powered materials in appropriate amounts of DMSO. The final concentration of DMSO never exceeded 5% (*v*/*v*) in reactions. The stock solution was stored at 5 °C until use.

#### 3.2.2. Cell Culture

Human epithelial lung carcinoma cells (A549) (ATCC^®^ CCL-185™, ATCC—Manassas, VA, USA) were grown in Kaighn’s Modification of Ham’s F-12 Medium (ATCC^®^ 30-2004™, ATCC—Manassas, VA, USA), and lung fibroblast cells (MRC5) (ATCC^®^ CCL-171™, ATCC—Manassas, VA, USA) were grown in Eagle’s Minimum Essential Medium (EMEM, ATCC^®^ 30-2003™, ATCC—Manassas, VA, USA). Ephitelial ovary adenocarcinoma cells (SKOV3) (ATCC^®^ HTB-77™, ATCC—Manassas, VA, USA) were grown in McCoy’s 5A medium (ATCC^®^ 30-2007™, ATCC—Manassas, VA, USA), and colon carcinoma cells (RKO) (ATCC^®^ CRL-2577™, ATCC—Manassas, VA, USA) were grown in Eagle’s Minimum Essential Medium (EMEM, ATCC^®^ 30-2003™, ATCC—Manassas, VA, USA). All were supplemented with 10% fetal bovine serum (FBS) (Sigma-Aldrich, Madrid, Spain) and 1% NORMOCIN solution (Thermo Fisher, Waltham, MA, USA). Cells were incubated at 37 °C in a 5% CO_2_ atmosphere and were split every 3–4 days to maintain monolayer coverage. For the cytotoxicity experiments, A549 cells were seeded in 96-well plates at a density of 2.5–3 × 10^3^ cells per well and incubated overnight to achieve 70% confluence at the time of exposure to the cytotoxic compound.

#### 3.2.3. Cytotoxicity Assays

Cells were incubated with different concentrations (50 µM, 30 µM, 20 µM, 10 µM, 5 µM, 2.5 µM and 1 µM) of the cytotoxic compounds for 48 h. Then, 10 µL of cell counting-kit 8 was added into each well, and cells were incubated for an additional 2 h at 37 °C in a 5% CO_2_ atmosphere. The absorbance of each well was determined by an Automatic Elisa Reader System (Thermo Scientific Multiskan FC, Thermo Scientific, Shangai, China) at a wavelength of 450 nm. If the obtained IC_50_ value was below 1 µM, the assay was repeated at lower concentrations (10 µM, 5 µM, 2.5 µM, 1 µM, 0.5 µM, 0.2 µM and 0.1 µM).

#### 3.2.4. Evaluation of Cytotoxicity Mechanisms

Flow cytometry assays were conducted using a FACSCalibur system flow cytometer (Becton Dickinson Bioscience, San Jose, CA, USA) in order to identify apoptotic cells and differentiate them from necrotic cells. A549 cells were exposed to 1 µM and 5 µM of cytotoxic compounds **4l** and **12a**, and cell apoptosis and necrosis were evaluated 24 h after exposure. For that purpose, treated cells were washed with phosphate buffered saline (PBS) (Sigma-Aldrich, Madrid, Spain) and detached with trypsin/EDTA (0.25%) (Gibco, Waltham, MA, USA). Cells were centrifuged at 1100 rpm for 5 min, and then the resulting pellet was resuspended in cell growth media and transferred to specific flow cytometer tubes. Propidium iodide (Sigma-Aldrich, Madrid, Spain) at a dilution of 1:300 was used in each sample to detect necrotic cells, and eBioscience™ Annexin V Apoptosis Detection Kit FITC (Fisher Scientific, Madrid, Spain) was used to detect apoptotic cells in accordance with the manufacturer’s instructions. The fluorescent signals corresponding to necrotic cells and apoptotic cells were measured at 650 nm (FL3) and 525 nm (FL1), respectively. Nontreated cells, used as negative control samples, were displayed on a forward scatter (FSC) versus side scatter (SSC) dot plot to establish a collection gate and exclude cells debris. Cells treated with 1 µM of camptothecin (Sigma-Aldrich, Madrid, Spain) served as a positive control for apoptosis and were used to establish cytometer settings and channel compensations. The experiments were carried out in triplicate for each condition. For each sample, 10,000 events were collected.

#### 3.2.5. Visualization of Cell Growth and Morphology

A qualitative analysis of A549 cell growth and morphology after exposure to 1.1 µM, 0.11 µM and 0.02 µM of the cytotoxic compound **4l** was conducted using the Cytation^TM^ 1 Cell Imaging Multi-Mode Reader (Biotek, Winooski, VT, USA). Cell images were acquired immediately after the addition of the compound and at the following time points after exposure: 1, 6, 12, 24, 48 and 72 h.

## 4. Conclusions

In conclusion, two complementary multicomponent methodologies were used for the preparation of 3-amino and 3-hydroxy 3-pyrrolin-2-ones, holding a variety of substituents at the five-membered ring. This strategy allows the possibility of structural diversity in the resultant scaffold depending on the starting amine, aldehyde and pyruvate or acetylene derivative, in a single step. The obtained γ-lactam derivatives showed in vitro cytotoxicity, inhibiting the growth of human tumor cells RKO (human colon epithelial carcinoma), SKOV3 (human ovarian carcinoma) and A549 (carcinomic human alveolar basal epithelial cell), and producing low activity toward MRC5 nonmalignant lung fibroblasts. QSAR studies indicate that the cytotoxicity is enhanced, in general, by the presence of aromatic groups bearing lipophilic methyl substituents or fluorine atoms. Moreover, the presence of an ester group at C-4 of the 5-membered heterocycle provided lower IC_50_ values than those previously reported for unsubstituted γ-lactam derivatives. Better antiproliferative activity was obtained for small methyl esters compared with ethyl or *iso*-propyl esters, which suggests a crucial interaction of the ester group, which can be complicated if a bulky group is present. In addition, while increased cytotoxicity was observed for 3-amino substituted γ-lactams compared with the 3-hydroxy substituted derivatives, the antiproliferative activity of such enol derivatives was enhanced when the flat ester group was isosterically replaced by a tetrahedral phosphonate or phosphine oxide moiety. Methyl ester-substituted γ-lactam **4e**, derived from *p*-toluidine, presented an excellent IC_50_ value of 1.67 µM against the A549 cell line with excellent selectivity towards the SKOV3 and RKO cell lines and nonmalignant cells. Its amide derivative **8a** showed good toxicity against lung, ovarian and colon cancer cells with IC_50_ values of 2.97, 6.95 and 18.67 µM, respectively, as well as very high selectivity towards malignant cell lines compared with noncancerous MRC5. The combination of phenol and methoxy moieties provided excellent IC_50_ values of 0.11 and 1.23 µM in the A549 and SKOV3 cell lines for *p*-toluidine-derived 3-amino γ-lactam **4l** with a 10 to 100 times higher selectivity level towards nonmalignant cells. The best level of cytotoxicity for 3-hydroxyγ-lactams was found for compound **4l** with an IC_50_ value of 4.00 μm in the A549 cell line. Likewise, the bioisosterism approach was shown to be an excellent tool in this study, leading to increased cytotoxicity for phosphorus-substituted γ-lactams with respect to the parent carboxylate-substituted derivatives with IC_50_ values of 1.46 and 21.97 µM in the A549 and SKOV3 cell lines in phosphine oxide substituted γ-lactam **12o**. These results may support the relevance of the isosterical substitution of carboxylic groups with tetrahedral phosphorus derivatives in view of their ability to block enzymes involved in the hydrolysis of peptides. In general terms, the γ-lactam derivatives described in this study were shown to be highly active towards the A549 cell line, while the SKOV3 and RKO cell lines were found to be more resistant. Importantly, most of the substrates showed high selectivity in cancer cells compared with nonmalignant cells. In addition, the cell morphology analysis and flow cytometry assays indicate that the main pathway by which γ-lactam derivatives induce cytotoxicity towards cancer cells is based on the activation of intracellular apoptotic mechanisms.

## Data Availability

Data is contained within the article and Appendix A.

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
