# Peer review of "Multicomponent Synthesis of Unsaturated γ-Lactam Derivatives. Applications as Antiproliferative Agents through the Bioisosterism Approach: Carbonyl vs. Phosphoryl Group"

_pharmaceuticals, 2022, doi:10.3390/ph15050511_

Round 1

Reviewer 1 Report

The paper submitted by J.-L. Pedraz, J. Vicario and co-workers describes an efficient multicomponent synthesis for the preparation of unsaturated γ-lactam derivatives (39 examples). The densely substituted γ-lactam substrates show 15 in vitro cytotoxicity, inhibiting the growth of carcinoma human tumor cell lines RKO (human colon 16 epithelial carcinoma), SKOV3 (human ovarian carcinoma) and A549 (carcinomic human alveolar 17 basal epithelial cell).

This work does not satisfy the criteria for originality, broad interest to a reasonable number of organic/pharmaceutical chemists to warrant disclosure as an article. In fact, the chemistry here reported is not novel and a number of articles involving a three-component reaction of amines, aldehydes and alkynes/pyruvates have already been published (for examples on the use of alkynoates, see: T. Liu, C. Dai, H. Sang, F. Chen, Y. Huang, H. Liao, S. Liu, Q. Zhu, J. Yang, Eur. J. Med. Chem. 2020, 199, 112334; D. Yang, C. Huang, H. Liao, H. Zhang, S. Wu, Q. Zhu, Z.-Z. Zhou, ACS Omega, 2019, 4, 17556−17560; M. M. Khan, S. Khan, Saigal, S. C. Sahoo, ChemistrySelect, 2018, 3, 1371–1380; L. Lv, S. Zheng, X. Cai, Z. Chen, Q. Zhu, S. Liu, ACS Comb. Sci. 2013, 15, 183−192; Q. Zhu, L. Gao, Z. Chen, S. Zheng, H. Shu, J. Li, H. Jiang, S. Liu, Eur. J. Med. Chem. 2012, 54, 232–238; for examples on the use of pyruvates, see the works by the same authors: del Corte, X.; Martinez de Marigorta, E.; Palacios, F.; Vicario, J., Molecules, 2019, 24, 2951–2962; del Corte, X.; López-Francés, A.; Maestro, A.; Martinez de Marigorta, E.; Palacios, F.; Vicario, J., J. Org. Chem. 2020, 85, 14369−14383). As a result, the majority of the reported compounds are not new; in fact, only 8 (4fm) out of 39 of these do not appear in the main chemistry databases such as SciFinder, and Belstein.

Moreover, the biological part is the copy of that reported in a previous published work by the same authors (Pharmaceuticals, 2021, 14, 782). Again, same cancer cell line such as A549, SKOV3 and MRC5, same studies on apoptosis and cell morphology.

In summary, the present work does not present enough novelty to justify its publication in a journal of high impact such as MDPI - Pharmaceuticals.

Reviewer 2 Report

The article “Multicomponent Syntheses of Unsaturated γ-Lactam Derivatives. Applications as Antiproliferative Agents through the Bioisosterism Approach: Carbonyl vs Phosphoryl Group” reports the synthesis of 3-amino and 3-hydroxy unsaturated γ-lactams in one step through a multicomponent reaction of amines, aldehydes and acetylene or pyruvate derivatives. The antiproliferative effects of the synthesized compounds was also evaluated.

 The chemistry part is well explained, only small comments and suggestions:

Suggestion: The authors could change Chart by Figure.

Page 4, line 145: I would consider low yield (< 35%), modest (35-50%), good yield (51-70%) and very good yield (> 70%).

Page 4, line 146: Please correct, scheme 5 to scheme 1.

Page 13, line 473: Please put at bold the compounds numbers.

Biological part has some issues, and the authors should revise this part with caution.

Please, correct IC50 to IC50 in tables.

Please, specify n.d. in table 3.

The results of the RKO cell line cytotoxic test should be present, at least in the SI.

What was the criterion for choosing 4l (1 µM) and 12a (5 µM) concentrations in A549 cells to clarify the mechanism of action?

Page 15, line 558: The authors write “The final concentration of DMSO never exceeded 5% (v/v) in reactions”. It is mean that the authors prepare a solution containing DMSO and other solvent? Which solvent?

Page 15, line 577: The authors should describe/specify “different concentrations” used to determine the IC50.

The conclusions are not consistent with the results presented. Must be reorganized and focused on results data.

Reviewer 3 Report

Vicario, Pedraz and coauthors report a comprehensive pharmacological study on 3-amino and 3-hydroxy 3-pyrrolin-2-ones, which turn out to be antiproliferative agents that operate on early apoptotic mechanisms. Without being a medicinal chemist I consider the biological investigations quite conclusive that also show a low toxicity of the compounds against non-malignal cell lines. Since the work has been submitted to a pharmaceutical journal the obtained data can be considered valuable for developing suitable API based upon the presented lead structure.

However, the underlying organic chemisty, although an elegant multicomponent approach was successfully applied to provide a large compound library, is not that novel and also previously communicated (Chart 1 contains 6 and Chart 3 contains 3 out of 7 compounds, previously reported compounds from Molecules 2019, 24, 2951, which is OK) or reported also with respect to its scope (Chart 2, all compounds were previously published, see JOC 2020, 85, 14369). To my opinion the syntheses appear some repeated and not necessarily novel (certainly in Chart 2, Chart 1/3 represent a logical extension of the methodology). This is not a big issue, however, I recommend restricting the the information to the presentation of then compounds in charts, without length scope discussions (which have already been performed, JOC 2020). The manuscript can be thereby shortened, referring to synthetic details in the Supp Inf.

In conclusion, this interesting work is worth publishing in Pharmaceuticals more from its medchem aspects than from its synthetic relevance and significance. The authors certainly have correctly referenced their own work. After having addressed this above mentioned minor issue the manuscript should be accepted for publication.

Round 2

Reviewer 1 Report

Although I am not entirely convinced whether the manuscript would be of great interest due to the lack of novelty on the syntheses of unsaturated gamma-lactam derivatives, I also must admit that the reported antiproliferative activity of all the new compounds could justify the publication in a journal like Pharmaceuticals.

Reviewer 2 Report

Multicomponent Syntheses of Unsaturated γ-Lactam Derivatives. Applications as antiproliferative Agents through the Bi-oisosterism Approach: Carbonyl vs Phosphoryl Group should be accepted at the present form.